# The Study of Correlation between Serum Vitamin D_3_ Concentrations and HBV DNA Levels and Immune Response in Chronic Hepatitis Patients

**DOI:** 10.3390/nu12041114

**Published:** 2020-04-16

**Authors:** Wang-Sheng Ko, Yen-Ping Yang, Fang-Ping Shen, Mu-Chen Wu, Chia-Ju Shih, Mei-Chun Lu, Yuan-Horng Yan, Ya-Ling Chiou

**Affiliations:** 1Department of Nutrition, Master of Biomedical Nutrition Program, Hungkuang University, Taichung 433, Taiwan; ker200448@gmail.com (W.-S.K.); a0939631309@gmail.com (F.-P.S.); sgazo063@hotmail.com (C.-J.S.); 2Department of Medicine Research, Kuang-Tien General Hospital, Taichung 433, Taiwan; meichunlu@gmail.com; 3Department of Internal Medicine, Kuang-Tien General Hospital, Taichung 433, Taiwan; 4Personnel Management Office, Ton-Yen General Hospital, Hsinchu 300, Taiwan; sheguorong@gmail.com; 5Department of Health Business Administration, Hungkuang University, Taichung 433, Taiwan; janice@sunrise.hk.edu.tw

**Keywords:** chronic hepatitis B, 25-hydroxyvitamin D_3_, HBV DNA, lymphocyte surface markers

## Abstract

Chronic hepatitis B (CHB) is a common chronic disease. Previous studies have shown a link between 25-hydroxyvitamin D_3_ (vitamin D_3_) concentration and liver disease. Hepatitis B virus (HBV) infection has been attributed to the inappropriate functioning of cell-mediated immunity. However, the effects of vitamin D_3_, immune cell, and HBeAg status on HBV viral load in CHB patients are still unclear. We investigated the relationship between the serum concentration of vitamin D_3_, percentage of immune cells in peripheral blood, and the HBV viral load of CHB patients. Sixty CHB patients were recruited, and their blood samples were collected and analyzed. Vitamin D level was measured using a chemiluminescence assay. A level of 30 ng/mL or above was defined as a vitamin D_3_ sufficiency. We assigned vitamin D_3_ status as either normal (≥30 ng/mL), insufficient (20–30 ng/mL), or deficient (<20 ng/mL). T-lymphocyte and B-lymphocyte surface markers in peripheral blood were detected using flow cytometry. The factors associated with HBV viral load were analyzed using univariate and multivariate-adjusted models. The mean serum vitamin D_3_ concentration in the subjects was 20.9 ± 5.6 ng/mL. Up to 88.3% of the patients were either deficient in or had insufficient vitamin D_3_. The gender, BMI, hepatitis B surface antigen levels, and ALT levels were significantly related to serum vitamin D_3_ levels. Serum vitamin D_3_ concentration, HBe status, HBs levels, ALT, and AST levels showed a statistically significant correlation with the HBV DNA levels. Serum vitamin D_3_ concentrations and hepatitis B surface antigen levels were strongly correlated with HBV DNA levels. Vitamin D_3_ levels were significantly associated with CD19 numbers (β:−6.2, 95% CI: −10.5). In multivariate analysis, vitamin D_3_ levels in the deficient and insufficient groups, and the CD8, HBeAg, and WBC counts were significantly associated with HBV DNA levels. In the immune tolerance phase of HBeAg-negative chronic HBV infection, vitamin D_3_ may be a modulator of immune function via CD8, CD19, and HBV DNA.

## 1. Introduction

Chronic hepatitis B (CHB), caused by infection by hepatitis B virus (HBV), is a major medical and public health issue worldwide. Roughly 30% of the world’s population shows serological evidence of current or past infection. CHB carriers are a high-risk group for developing cirrhosis and hepatocellular carcinoma. Approximately 50% of CHB infected patients die from liver cancer or cirrhosis in Taiwan [1]. HBV is a partially double-stranded DNA virus with several serological markers: HBsAg, anti-HBs, HBeAg, anti-HBe, anti-HBc, IgM, and IgG [2]. HBV DNA and HBeAg are used as markers of viral replication in CHB patients [3]. The main reasons CHB infections cannot be cured at present are associated with the host gene, the degree of immunity, the genotype and variant of the virus, the amount of virus, and other yet unknown causes. The lower the level of HBV before treatment, the higher the liver inflammation index. Consequently, this leads to a lower level of HBs surface antigen and a higher response rate to treatment [4].

Patients infected with HBV can develop both an innate and an adaptive immune response. HBV infected liver cells secrete interferon γ, α/β and tumor necrosis factor α which inhibit HBV replication. HBV infected liver cells can activate natural killer cells and natural killer T cells (NKT cells) followed by T cells (predominantly cytotoxic T cells) and B cells [5]. Previous studies have shown that the proportion of CD4^+^ T cells in the blood of patients with CHB will decrease but the proportion of CD8^+^ T cells will increase. The proportion of CD8^+^ T cells is inversely proportional to the serum HBV concentration [6]. In addition, previous studies have shown that HBV affects host immunity not only in CD4^+^ and CD8^+^ T cells, but also in the number of CD45RO^+^ T cells (memory T cells) [7,8]. CD45RA and CD45RO are markers at different stages of lymphoblast differentiation. CD45RA is expressed on the surface of naive T cells and CD45RO is expressed on the surface of memory T cells. Activated CD45RO^+^ T cells release many lymphokines to trigger or maintain an immune response.

The most common functions of vitamin D are to stimulate the absorption of calcium and phosphorus in the gastrointestinal tract and increase bone mineral content and remodeling. It is currently considered to be a steroid hormone and organs and tissues, including pancreas, large intestine, small intestine, muscles and nervous system, with vitamin D receptors can all be regulated [9]. Currently, vitamin D receptors are widely found in systemic organs and tissues including bone, the immune system, brain, cardiac system and thyroid. Research shows that vitamin D can produce extra-skeletal effects such as inhibiting cancer progression, skin repair, protecting the cardiovascular system, regulating innate and acquired immunity, eliminating viruses and bacteria and treating and preventing autoimmune diseases. Studies have shown that the progression and prognosis of liver disease is correlated to serum vitamin D concentrations [10]. Vitamin D supplementation for autoimmune liver disease has been shown to have therapeutic effects [11,12]. Vitamin D is mainly composed of 25 (OH) D_3_ (vitamin D_3_) in the blood. There is currently no consensus on the standard of vitamin D_3_ deficiency and deficiency in skeletal and non-skeletal functions. The Institute of Medicine of the National Academies (IOM) recommends that the concentration of vitamin D_3_ is greater than 20 ng/mL. The Endocrine Society and International Osteoporosis Foundation (IOF) recommend the concentration of 25 (OH) D3. More than 30 ng/mL is sufficient [13]. According to the above definition, about 1 billion people in the world are deficient in vitamin D [14]. Clinical and epidemiological data indicate that chronic viral hepatitis is related to vitamin D concentration [15]. In hepatitis diseases, the serum vitamin D_3_ concentration of patients with chronic hepatitis C was lower than that of healthy people, and the concentration of vitamin D_3_ in patients with chronic hepatitis C was less than 30 ng/mL (73%). Patients with severe hepatitis C fibrosis have lower vitamin D concentrations than mild fibrosis [16]. The prevalence of vitamin D_3_ deficiency (<20 ng/mL) in patients with CHB is as high as 81%. Patients with chronic hepatitis B have lower vitamin D_3_ levels compared to patients with chronic hepatitis C. Vitamin D is an independent factor for the level of hepatitis B virus and serum vitamin D concentration is negatively correlated with serum HBV DNA levels. The serum vitamin D_3_ concentration and the level of HBV DNA are impacted by the season. In the autumn and winter seasons, patients have a lower vitamin D concentration and higher levels of HBV DNA compared to the summer and spring seasons [17]. Some studies have shown that HBV reduces the expression of vitamin D receptors in HBV-infected cells, reduces vitamin D from aiding the immune defense system, and thus reduces the effect of inhibiting viral replication [18,19]. This indicates that vitamin D_3_ affects the immune status of patients by influencing the level of HBV DNA. In turn, the level of HBV DNA influences the efficacy of treatment of hepatitis B [20,21].

Furthermore, the chronicity of hepatitis B virus infection has been attributed to the inappropriate functioning of cell-mediated immunity. However, the influences of vitamin D_3_, immune cell and HBeAg status on HBV viral load in CHB patients is still unclear and has not been reported for Taiwan. The objective of this study was to investigate the relationship between serum concentrations of vitamin D, percentage of immune cells in peripheral blood and HBV viral load in CHB patients.

## 2. Materials and Methods

### 2.1. Study Participants Kuang-Tien

The patients were enrolled from among the outpatients of General Hospital from May to September 2016. The following were used as the inclusion criteria: chronic hepatitis B carriers, HBeAg-positive or -negative, and ALT less than two times that of the normal value in healthy individuals (>40 units/L and <80 units/L). The exclusion criteria diagnosed by a physician were cirrhosis, liver cancer, hepatitis C, diabetes, immune diseases, other malignancies, other chronic diseases, infection, pregnancy, or other systematic immune diseases. A total of 60 CHB patients were enrolled from a Liver Disease Clinic within the hospital located in the mid-coastal region of Taiwan. The study protocol was approved by the ethics committees (KTGH IRB No: 10501), and informed consent was obtained from all study participants. The study was carried out in accordance with the ethics principles of the Declaration of Helsinki and was consistent with good clinical practice guidelines and local regulatory requirements.

### 2.2. Analysis of Basic and Biochemical Data of Subjects

Demographic data were collected including age, gender, education level, smoking, alcohol drinking, nutrition supplementation, and body mass index (BMI). Blood testing was performed including serum vitamin D_3_ concentrations, blood cell analysis, biochemical tests and HBV DNA levels. Heparinized blood was collected to evaluate liver and kidney function and genotype, and to quantify HBV DNA concentration, HBsAg, anti-HBs, HBeAg, and anti-HBeAg. The remaining blood was segregated into serum and cells. Serum was stored as aliquots in liquid nitrogen until analysis and cells were analyzed to determine the T cell subtypes. Complete blood count (CBC) examinations, including hemoglobin (Hb), white blood cell (WBC) count, neutrophil, lymphocyte, and platelet (PLT), were performed using a Sysmex XE-5000 hematology analyzer (Sysmex Corporation, Kobe, Japan). Total cholesterol, triglyceride, creatinine, and fasting glucose were measured as part of the routine serum biochemistry profile (Beckman Coulter, California, USA). Vitamin D level was measured using chemiluminescence assay. HBsAg level was measured using Chemiluminescent microparticle immunoassay by Architect i2000SR analyzer (Abbott Diagnostic, Chicago, IL, USA). HBV DNA loads were measured by real time polymerase chain reaction (RT-PCR) using the 7500 Fast Real-Time PCR System (Applied Biosystems, California, USA). Liver function tests, including assessment of alanine transaminase (ALT) and aspartate transaminase (AST) enzyme levels, platelet count (PLT), total and direct bilirubin, and albumin, were performed using a Beckman UniCel DxC800 Synchron (Analyzer (Beckman Coulter, California, USA). Total HBsAg, HBeAg, and anti-HBe were determined using Radio-immunoassay Kit and by Elecsys 2010 system (Roche, Mannheim, Germany).

### 2.3. Analysis of T Cell Subtypes

The cells were incubated with fluorescein isothiocyanate (FITC)-conjugated anti-human CD3, anti-human CD25, anti-human CD45RO, or anti-human CD45RO antibodies for 20 min at 4 °C in the dark, followed by staining with phycoerythrin (PE)-conjugated anti-human CD4, anti-human CD8, or CD19 antibodies, for separating individual T cell subtypes. PE-conjugated mouse IgG1 antibodies were used as isotype controls. These antibodies were purchased from BD Company (Franklin Lakes, NJ, USA). T-lymphocyte surface makers (CD3^+^CD4^+^, CD3^+^CD8^+,^ CD4^+^CD45RA^+^, CD4^+^CD45RO^+,^ CD8^+^CD45RA^+^, CD8^+^CD45RO^+^), and B-lymphocyte surface makers (CD19^+,^ CD19^+^CD45RA^+^, CD19^+^CD45RO^+^) in peripheral blood were detected using flow cytometry analysis.

### 2.4. Statistics

Continuous data with normal distribution were presented as mean ± SD and performed by ANOVA; non-normal distribution data were presented as median (IQR) using the Kruskal-Wallis test. Categorical data were presented as *n* (%) using the chi-squared test or Fisher’s exact test as appropriate. Correlations between variables were evaluated using the Pearson’s correlation test or Spearman’s rank correlation test as data with or without normal distribution. After univariate analyses, multivariate analyses were performed for significant associations. Multivariate models were obtained by backward selection, using *p* value > 0.15 for removal from the model. Only patients with complete data for the remaining covariates were included in multivariate analyses. A two-sided *p*-value < 0.05 was considered statistically significant. All analyses were performed using SAS 9.4 (SAS Institute, Cary, NC, USA).

## 3. Results

### 3.1. Characteristics of Patients

In this study, CHB patients had a mean serum vitamin D_3_ concentration of 20.9 ± 5.6 ng/mL. We categorized the vitamin D_3_ status as normal (≥30 ng/mL), insufficient (20–30 ng/mL) and deficient (<20 ng/mL). Up to 88.3% of patients were either vitamin D_3_ deficient or insufficient. The gender and body mass index (BMI) significantly correlated to serum vitamin D levels in these CHB patients (*p* = 0.04) (Table 1).

### 3.2. The Connection Between Hbeag Levels and Vitamin D3 Status

The relationships between vitamin D_3_ status and various laboratory parameters are shown in Table 2. 89.7% of study participants were HBeAg-negative. Hepatitis B surface antigen (HBsAg) levels are significantly correlated to serum vitamin D_3_ levels in CHB patients (*p* = 0.00). The lower levels of HBeAg were in the normal (≥30 ng/mL) and insufficient (20–30 ng/mL) vitamin D_3_ group. The levels of Hemoglobin (Hb), PLT, Alanine aminotransferase (ALT) and total cholesterol also showed similar trends.

### 3.3. The Connection Between Immunity and Vitamin D3 Status

Table 3 shows the relationship between vitamin D and levels of peripheral immune cells. Study participants in the 3 vitamin D groups had CD19^+^ cell counts of 3.1 ± 2.5, 10.9 ± 5.2, and 8.0 ± 4.1 (%), respectively (*p* = 0.001).

### 3.4. Factors Associated With HBV DNA Serum Concentration

Table 4 shows the factors associated with HBV DNA serum concentration (log10 IU/mL) using univariate models. Serum vitamin D_3_ concentrations, HBeAg status, HBsAg levels, WBC count, and AST showed statistical significance with HBV DNA levels. In T-lymphocyte surface markers, the proportion of CD4^+^, CD8^+^, CD45^+^ cells showed statistical significance with HBV DNA levels. In B-lymphocyte surface markers, the proportion of CD19^+^ cells showed statistical significance with HBV DNA levels.

### 3.5. Estimates for HBV DNA Serum Concentration Using Multivariate-Adjusted Models

Table 5 shows the results of multivariate-adjusted models. Levels of vitamin D in the deficient and insufficient range, CD8, HBeAg status, and WBC counts were significantly associated with HBV DNA levels (β:1.28, 95% CI:0.17, 2.39; β:1.99, 95% CI:0.88, 3.10; β:0.09, 95% CI:0.04, 0.14; β:−4.1. 95% CI:−5.28,−2.93; β:−0.28, 95% CI:−0.52, −0.05; all *p* < 0.05).

## 4. Discussion

Our results showed that at the vitamin D_3_ deficient and insufficient concentrations, CD8 levels, HBeAg status and WBC counts were significantly associated with HBV DNA levels. This suggests that in the immune tolerance phase of HBeAg-negative chronic HBV infection, vitamin D_3_ may be a modulator of immune function via CD8, CD19 and white blood cells. The results provide supporting evidence that vitamin D_3_ has roles in immune modulation and regulation of inflammation in CHB patients in the immune tolerance phase. We were also first to find out that vitamin D_3_ levels correlate with the levels of lymphocyte subsets, CD8, CD19, in patients with liver disease.

Previous studies have described the factors that influence vitamin D_3_ levels and vitamin D_3_ deficiency. Hoan et al. examined the correlation between vitamin D levels and HBV-DNA loads in 165 CHB patients. They found that vitamin D levels and HBV-DNA were significantly and inversely correlated. Investigation of PLT, ALT, AST, total-bilirubin, direct-bilirubin, albumin, AFP and prothrombin levels showed that they were either weakly correlated or did not correlate with vitamin D_3_ serum levels. Multivariate logistic regression models showed that HBeAg was negatively and independently associated with low vitamin D levels [22]. In another study, Chan and colleagues enrolled 737 CHB patients and found that 35% had insufficient and 58% had deficient vitamin D_3_ levels. Using univariate analyses, lower vitamin D_3_ levels were found to be significantly associated with age, lower uric acid levels, HBeAg-positive status, lower calcium levels, season of blood draw (winter or autumn) and HBV genotype D. Using multivariate analyses, only HBV genotype, season of blood draw, calcium level and age showed an association [23].

One recent study reported correlations between vitamin D_3_ levels and the lymphocyte subsets in 8621 elderly patients with age-related diseases including hypertension, cardiovascular disease (CRVD), cerebrovascular disease (CAD) and type 2 diabetes mellitus (T2DM). More than 70% of the patients in each disease group had total a vitamin D concentration of < 20 ng/mL. In CRVD patients, CD3 and CD19 correlated with 25(OH) vitamin D_3_. In CAD patients, CD8, CD4, CD19 and CD4/CD8 correlated with 25(OH) vitamin D_2_ and CD8 correlated with 25(OH) vitamin D_3_. In T2DM and hypertension patients, CD8, CD3, CD19 and CD4/CD8 correlated with 25(OH) vitamin D_3_. Though there were some correlations between 25(OH) vitamin D_2_ and 25(OH) vitamin D_3_ status and lymphocyte subsets, the results for kidney disease, digestive disease and fracture/osteoporosis groups were insufficient to draw an inference on the effects of serum levels of vitamin D metabolites on these diseases [24]. We found that vitamin D correlated with the number of CD8 cells, which is consistent with the results of the above studies. In addition, we found that CD19 may be an important lymphocyte subset between vitamin D_3_ and CHB.

CD19 impacts B lymphocyte activation and differentiation through the modulation of B cell receptor signaling. CD19 is important for establishing optimized immune responses by modulating antigen-independent B cell development and immunoglobulin-induced B lymphocyte activation [25]. Previous studies have shown a relationship between CD19 and hepatitis C, but few studies of CD19 and CHB [26,27,28,29]. In patients with recurrent miscarriage, the percentage of CD19^+^ B cells was significantly higher in the vitamin D_3_ insufficiency group than in the vitamin D_3_ normal group [30]. In children with community-acquired pneumonia, the positive CD19^+^ cells in patients with vitamin D deficiency were significantly less than that in patients without vitamin D_3_ deficiency [31]. We first reported the evidence linking vitamin D level and CD19 in CHB patients. Further mechanistic studies are needed.

While the potential anti-viral effect of vitamin D_3_ has been described, the mechanisms remain unclear. The interplay between viral infections and vitamin D_3_ remains an intriguing concept. The overall effect vitamin D_3_ can have on the immune signature in the context of viral infections is an area of growing interest [32]. One recent study showed that severe vitamin D_3_ deficiency was associated with treatment non-response, progression to cirrhosis, and liver-related death. Severe vitamin D_3_ deficiency was a prognostic biomarker in autoimmune hepatitis (AIH) [33]. One human study reporting the predictive value of pre-treatment with vitamin D_3_ on the virologic response in treated CHB patients. Serum 25(OH) vitamin D levels were affected by gender, season, latitude, and genetic variation in vitamin D_3_ metabolism [34]. One animal study investigated the therapeutic effect of vitamin D_3_ combined with interferon on mice with hepatitis B. The percentage of CD4^+^ and the CD4^+^/CD8^+^ ratio was significantly increased, but the percentage of CD8^+^ was reduced. The proliferation rate of splenic lymphocytes was higher. Higher efficacy was achieved for the treatment of hepatitis B in mice, possibly through increasing the immune level of mice [35]. However, the effect of vitamin D_3_ supplementation in combination with IFN-RBV based therapy on the virologic response is still unclear. There is currently no approved recommendation for the use of vitamin D_3_ supplementation and vitamin D_3_ analogues as supportive adjuvant treatment regimens in viral hepatitis. Further randomized, placebo-armed studies need to be performed in order to confirm whether supplementation of vitamin D_3_ or vitamin D_3_ analogues improves sustained virologic responses (SVRs) in combination with specific antiviral treatment strategies in HBV infections [36].

## 5. Conclusions

In summary, we found that, in the immune tolerance phase of HBeAg-negative chronic HBV infection, vitamin D may be a modulator of immune function via CD8, CD19 and white blood cells. Future large-scale studies are likely to confirm our findings.

## Figures and Tables

**Table 1 nutrients-12-01114-t001:** Baseline characteristics of patients included in the study.

Variable	Number	Vitamin D, ng/mL			*P*
		<20 (*n* = 27)	20–30 (*n* = 26)	≥30 (*n* = 7)	
Age	60	44.0 ± 11.2	45.5 ± 10.1	52.4 ± 10.1	0.18 ^a^
BMI (kg/m^2^)	60	21.9 (20.2–25.7)	25.4 (23.0– 28.1)	23.1 (20.6– 23.7)	0.04 ^b^
Sex					0.04 ^c^
Male	35	11 (40.7)	18 (69.2)	6 (85.7)	
Female	25	16 (59.3)	8 (30.8)	1 (14.3)	
Education					0.89 ^c^
Under junior high school	6	3 (11.1)	3 (11.5)	0 (0.0)	
Senior high school	19	7 (25.9)	9 (34.6)	3 (42.9)	
Above college	35	17 (63.0)	14 (53.9)	4 (57.1)	
Nutrition					0.20 ^c^
No	35	18 (66.7)	15 (57.7)	2 (28.6)	
Yes	25	9 (33.3)	11 (42.3)	5 (71.4)	
Exercise per-week frequency					0.18 ^c^
No	26	14 (51.9)	10 (38.5)	2 (28.6)	
1	15	9 (33.3)	5 (19.2)	1 (14.3)	
1–3	9	2 (7.4)	6 (23.1)	1 (14.3)	
>3	10	2 (7.4)	5 (19.2)	3 (42.9)	
Smoke status					0.13 ^c^
No	55	26 (96.3)	24 (92.3)	6 (71.4)	
Yes	5	1 (3.7)	2 (7.7)	2 (28.6)	
Drink status					0.66 ^d^
No	45	19 (70.4)	21 (80.8)	5 (71.4)	
Yes	15	8 (29.6)	5 (19.2)	2 (28.6)	

^a^: Data were continuous and presented as mean ± SD using ANOVA; ^b^: Data were continuous and presented as median (IQR) using Kruskal-Wallis test; ^c^: Data were categorical and presented as *n* (%) using Fisher’s exact test; ^d^: Data were categorical and presented as *n* (%) using chi-squared test.

**Table 2 nutrients-12-01114-t002:** Laboratory data of patients included in the study.

Variable	Number	Vitamin D, ng/mL			*P*
		<20 (*n* = 27)	20–30 (*n* = 26)	≥30 (*n* = 7)	
HBeAg					0.47 ^c^
negative	54	25 (92.6)	22 (84.6)	7 (100)	
positive	6	2 (7.4)	4 (15.4)	0 (0)	
HBsAg, log10 IU/mL	60	3.0 (1.8–3.2)	3.2 (2.3–3.5)	0 (0–2.4)	0.002 ^b^
Hemoglobin, g/dL	60	13.6 ± 1.5	14.7 ± 1.2	14.0 ± 1.9	0.02 ^a^
WBC count, ×10^3^/µL	60	5.6 (4.5–6.5)	5.3 (4.8–6.2)	4.7 (4.0–6.5)	0.74 ^b^
Platelet, ×10^3^/µL	60	213.0 (192.0–261.0)	202.0 (178.0–244.0)	164.0 (137.0–209.0)	0.04 ^b^
Lymphocyte, %	60	35.9 ± 6.6	33.4 ± 6.1	29.5 ± 5.7	0.05 ^a^
Neutrophil, %	60	55.4 ± 7.0	59.0 ± 7.1	60.8 ± 6.0	0.08 ^a^
Total bilirubin, mg/dL	58	0.8 (0.7–1.0)	1.0 (0.8–1.1)	0.9 (0.7–1.2)	0.15 ^b^
Albumin, g/dL	58	4.7 (4.5–4.8)	4.9 (4.6–5.0)	4.6 (4.6–4.9)	0.08 ^b^
AC Sugar, g/dL	56	96.5 (94.0–102.0)	100.0 (96.0–105.0)	101.5 (98.0–105.0)	0.18 ^b^
ALT, IU/L	60	23.0 (18.0–30.0)	33.0 (26.0–49.0)	21.0 (16.0–24.0)	0.01 ^b^
AST, IU/L	60	23.0 (21.0–26.0)	26.5 (22.0–31.0)	23.0 (19.0–26.0)	0.26 ^b^
Triglyceride, mg/dL	60	91.0 (61.0–137.0)	87.0 (66.0–113.0)	59.0 (48.0–71.0)	0.09 ^b^
Cholesterol, mg/dL	60	180.0 (158.0–208.0)	205.5 (191.0–229.0)	171.0 (145.0–194.0)	0.02 ^b^
Creatinine, mg/dL	57	0.7 ± 0.2	0.9 ± 0.2	0.9 ± 0.2	0.04 ^a^

^a^: Data were continuous and presented as mean ± SD using ANOVA; ^b^: Data were continuous and presented as median (IQR) using Kruskal-Wallis test; ^c^: Data were categorical and presented as *n* (%) using Fisher’s exact test; ^d^: Data were categorical and presented as *n* (%) using chi-squared test.

**Table 3 nutrients-12-01114-t003:** T-lymphocyte and B-lymphocyte surface makers.

Variable	Number	Vitamin D			*P*
		<20	20–30	≥30	
T-lymphocyte surface markers					
CD3	60	44.4 ± 17.0	44.5 ± 14.0	41.4 ± 23.4	0.90 ^a^
CD4	60	13.5 ± 8.5	11.4 ± 6.2	11.1 ± 6.6	0.54 ^a^
CD3CD4	60	10.2 (6.2–15.2)	9.2 (5.6–13.8)	11.3 (5.8–16.4)	0.78 ^b^
CD8	60	10.9 ± 6.8	8.6 ± 5.9	6.9 ± 7.5	0.25 ^a^
CD3CD8	60	10.6 (3.9–13.4)	6.3 (3.1–9.5)	3.9 (0.9–9.3)	0.15 ^b^
CD45RA	60	28.6 (19.9–34.9)	31.3 (21.8–36.5)	14.3 (4.9–31.8)	0.26 ^b^
CD4 CD45RA	60	3.0 (1.3–4.7)	3.2 (2.2–4.6)	0.5 (0.1–4.1)	0.15 ^b^
CD45RO	60	7.3 (4.6–10.4)	8.1 (2.9–11.1)	9.6 (7.9–11.9)	0.58 ^b^
CD4 CD45RO	60	5.0 (2.3–8.8)	3.4 (1.4–6.1)	5.4 (4.3–9.4)	0.47 ^b^
CD25	60	0.5 (0.3–1.6)	0.5 (0.3–3.3)	0.6 (0.2–1.4)	0.97 ^b^
CD4CD25	60	0.2 (0.1–0.5)	0.2 (0.1–0.3)	0.1 (0.0–0.3)	0.40 ^b^
CD8 CD45RA	60	5.4 (2.2–8.5)	5.9 (2.7–8.1)	3.6 (0.3–8.3)	0.56 ^b^
CD8 CD45RO	60	1.4 (0.8–2.8)	1.0 (0.6–1.7)	1.1 (0.1–2.4)	0.26 ^b^
B-lymphocyte surface markers					
CD19	49	8.0 ± 4.1	10.9 ± 5.2	3.1 ± 2.5	0.001 ^a^
CD19 CD45RA	49	5.4 ± 2.6	7.5 ± 3.7	2.7 ± 2.4	0.004 ^a^
CD19 CD45RO	49	0.1 (0.0–0.1)	0.0 (0.0–0.1)	0.0 (0.0–0.6)	0.94 ^b^

^a^: Data with normal distribution were presented as mean ± SD and performed by ANOVA; ^b^: Data with non-normal distribution were presented as median (25–75) and performed by Kruskal-Wallis test.

**Table 4 nutrients-12-01114-t004:** Factors associated with HBV DNA serum concentration (log10 IU/mL).

Variable	Number	Median (IQR), Mean ± SD, or Correlation Coefficient	*P*
Vitamin D, ng/mL			0.03 ^b^
<20	27	3.0 (2.4–4.5)	
20–30	26	3.8 (2.4–5.6)	
≥30	7	0 (0–3.1)	
Baseline Characteristic			
Age	60	−0.08	0.56 ^c^
BMI	60	−0.16	0.23 ^d^
Gender			0.99 ^b^
Male	35	3.3 (1.7–5.0)	
Female	25	3.1 (2.4–4.1)	
Education			0.18 ^b^
Under junior high school	6	2.8 (2.6–3.5)	
Senior high school	19	4.8 (2.2–5.6)	
Above college	35	3.1 (1.8–4.5)	
Nutrition			0.15 ^b^
No	35	3.5 (2.4–5.0)	
Yes	25	2.8 (1.6–3.7)	
Exercise per-week frequency			0.88 ^a^
No	26	3.8 ± 2.0	
1	15	3.2 ± 2.0	
1–3	9	3.2 ± 2.6	
>3	10	3.5 ± 2.7	
Smoke status			0.16 ^b^
No	55	3.3 (2.4–4.8)	
Yes	5	1.5 (0–3.3)	
Drink status			0.74 ^b^
No	45	3.3 (2.4–4.8)	
Yes	15	2.7 (2.0–4.5)	
Laboratory data			
HBeAg			<0.001 ^b^
negative	54	3.1 (2.0–4.1)	
positive	6	8.2 (8.0–8.2)	
HBsAg, log10 IU/mL	60	0.60	<0.001 ^d^
Hemoglobin, g/dL	60	0.12	0.36 ^d^
WBC count, ×10^3^/µL	60	−0.35	0.01 ^d^
Platelet, ×10^3^/µL	60	−0.08	0.54 ^d^
Lymphocyte, %	60	0.08	0.53 ^c^
Neutrophil, %	60	−0.11	0.41 ^c^
Total bilirubin, mg/dL	58	0.24	0.07 ^d^
Albumin, g/dL	58	−0.21	0.12 ^d^
AC Sugar, g/dL	56	0.08	0.58 ^d^
GPT, IU/L	60	0.22	0.10 ^d^
GOP, IU/L	60	0.30	0.02 ^d^
Triglyceride, mg/dL	60	−0.15	0.26 ^d^
Cholesterol, mg/dL	60	0.17	0.19 ^c^
Creatinine, mg/dL	57	0.03	0.82 ^c^
T-lymphocyte surface makers			
CD3	60	0.28	0.03 ^c^
CD4	60	0.15	0.27 ^c^
CD3CD4	60	0.20	0.12 ^d^
CD8	60	0.41	0.00 ^d^
CD3CD8	60	0.39	0.00 ^d^
CD45RA	60	0.34	0.01 ^d^
CD4 CD45RA	60	0.33	0.01 ^d^
CD45RO	60	0.10	0.43 ^d^
CD4 CD45RO	60	0.16	0.23 ^d^
CD25	60	0.06	0.64 ^d^
CD4CD25	60	0.24	0.07 ^d^
CD8 CD45RA	60	0.40	0.00 ^d^
CD8 CD45RO	60	0.14	0.29 ^d^
B-lymphocyte surface markers			
CD19	49	0.33	0.02 ^d^
CD19 CD45RA	49	0.36	0.01 ^d^
CD19 CD45RO	49	−0.10	0.48 ^d^

^a^: Data were classified into more than 3 groups with normal distribution and presented as mean ± SD of HBV DNA (log10 IU/mL) using ANOVA; ^b^: Data were classified without normal distribution and presented as median (IQR) using Wilcoxon rank-sum test (2 groups) or Kruskal-Wallis test (more than 3 groups); ^c^: Data were continuous with normal distribution and presented as Pearson’s correlation coefficient; ^d^: Data were continuous without normal distribution and presented as Spearman’s correlation coefficient.

**Table 5 nutrients-12-01114-t005:** Estimates for HBV DNA serum concentration (log10 IU/mL) using multiple linear regression.

Variable	Estimate	95% CI	SE	*P*
Intercept	6.51	(4.69–8.32)	0.91	<0.0001
Vitamin D, ng/mL				
<20	1.28	(0.17–2.39)	0.55	0.025
20–30	1.99	(0.88–3.10)	0.55	0.001
≥30	reference			
CD8	0.09	(0.04–0.14)	0.03	0.001
HBeAg				
negative	−4.11	(−5.28–−2.93)	0.59	<0.0001
positive	reference			
WBC count, ×10^3^/µL	−0.28	(−0.52–−0.05)	0.12	0.018

CI, Confidence interval; SE, standard error.

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
