# Peer review of "The Study of Correlation between Serum Vitamin D3 Concentrations and HBV DNA Levels and Immune Response in Chronic Hepatitis Patients"

_nutrients, 2020, doi:10.3390/nu12041114_

Round 1
Reviewer 1 Report
In this study, Ko et. al studied 60 patients with chronic hepatitis B and correlated serum vitamin D levels to disease status (HBV DNA, HBeAg, HBsAg, etc.). The authors found statistically significant correlations between vitamin D levels and patient sex, HBV DNA, HBsAg, CD19+ cell count, and other measures. The manuscript is interesting, well-written, and contains valuable data. My main concern is the lack of depth in discussion of the data and relevant previous studies. Vitamin D has been correlated with many diseases, including infectious diseases, cancers, and autoimmune diseases. In most cases, however, a direct link between vitamin D level and disease state does not exist. Physiological measures are so variable across patients that establishing a link between variable X and disease Y is nearly impossible without controlled experimentation with proper treatment groups. In the case of vitamin D, most humans (not just CHB patients) likely fall into the vitamin D inefficient/deficient categories defined in this current manuscript. I believe the manuscript would be stronger and more informative if the data is connected to other studies. Some suggestions are below:
- The authors state that serum vitamin D and HBV DNA levels are affected by the season (ref 13). When were samples from patients taken for this study? Were they all taken at the same time in the same season, or were they from different times of the year? Please provide more details on sample collection.
- Vitamin D has been correlated to measures of HBV infection in previous studies. Indeed, the authors reference a previous study (ref 13) that found vitamin D is negatively correlated with HBV DNA level. What is new in this current manuscript? What is different than previously reported data? How does the data compare to previous data?
- Please provide detailed discussion of vitamin D status in non-HBV-infected individuals, any previous data on vitamin D and all stages of HBV infection, and any data available on vitamin D related to other hepatitis viruses.
- Vitamin D receptor (VDR) has an established role in immune function. Please discuss the potential for VDR activity and modulation in this CHB context.
- Lines 221-222: “We found that vitamin D affects the number of CD8 cells, which is consistent with the results of the above studies.” It was not found that vitamin D affects anything. Vitamin D levels were only correlated with other factors. Please rephrase.
Author Response
- The authors state that serum vitamin D and HBV DNA levels are affected by the season (ref 13). When were samples from patients taken for this study? Were they all taken at the same time in the same season, or were they from different times of the year? Please provide more details on sample collection.
A: Thanks for the reviewer ’s suggestion. The patients were enrolled from among the outpatients of Kuang-Tien General Hospital from May to September 2016 (p2).
- Vitamin D has been correlated to measures of HBV infection in previous studies. Indeed, the authors reference a previous study (ref 13) that found vitamin D is negatively correlated with HBV DNA level. What is new in this current manuscript? What is different than previously reported data? How does the data compare to previous data?
A: Thanks for the reviewer ’s suggestion. This study is based on data from Taiwan, and it is found that in sunny Taiwan, as in other countries, there is the problem of vitamin D3 deficiency in HBV infection patients. This study also explores the lack of vitamin D3 in hepatitis B patients with the proportion of immune cells in the immune cell changes, we also found In the immune tolerance phase of HBeAg-negative chronic HBV infection, vitamin D3 may be a modulator of immune function via CD8, CD19, and HBV DNA.
- Please provide detailed discussion of vitamin D status in non-HBV-infected individuals, any previous data on vitamin D and all stages of HBV infection, and any data available on vitamin D related to other hepatitis viruses.
A: Thanks for the reviewer ’s suggestion, we have revised the content in the manuscript (Lines 71-83)
- Vitamin D receptor (VDR) has an established role in immune function. Please discuss the potential for VDR activity and modulation in this CHB context.
A: Thanks for the reviewer ’s suggestion, we have revised the content in the manuscript (Lines 89-91). Some studies have shown that HBV reduces the expression of vitamin D receptors in HBV-infected cells, prevents vitamin D from causing an immune defense system, and thus reduces the effect of inhibiting viral replication.
- Lines 221-222: “We found that vitamin D affects the number of CD8 cells, which is consistent with the results of the above studies.” It was not found that vitamin D affects anything. Vitamin D levels were only correlated with other factors. Please rephrase.
A: Thanks for the reviewer ’s suggestion, we have revised the content in the manuscript (Lines 221-222). We found that vitamin D correlated with the number of CD8 cells, which is consistent with the results of the above studies.

Reviewer 2 Report
The study reports a link between vitamin D deficiency/insufficiency with higher serum HBV DNA/antigen and CD8/CD19 subset levels. Given that there is a known negative correlation between HBV antigen levels and the functionality of HBV-specific CD8 T-cells, this reports highlights the importance of vitamin D and their roles in modulating immune functions in CHB patients. A further study in the future to analyse the role of Vitamin D deficiency/insufficiency in establishing chronic hepatitis B infection, using a surrogate animal model for CHB can provide a clear understanding to the significance of the findings from this work.
The results provided in this work is well presented and has significant impact for the scientific community.
Minor corrections/suggestions:
- Table 2: (i) red colour highlights can be consistent across rows of all important blood parameters (ii) GOT, GPT can be interchanged or given in brackets for AST and AST, respectively.
- Table 3: layout of data for Vitamin D levels has been reversed (normal to insufficient/deficient), this can be presented from insufficient/deficient to normal, to be consistent with data in tables 1 and 2.
Author Response
1.Table 2: (i) red colour highlights can be consistent across rows of all important blood parameters (ii) GOT, GPT can be interchanged or given in brackets for AST and AST, respectively.
A: Thanks for the reviewer ’s suggestion, we have revised the content in the manuscript (p4 and p5)
2.Table 3: layout of data for Vitamin D levels has been reversed (normal to insufficient/deficient), this can be presented from insufficient/deficient to normal, to be consistent with data in tables 1 and 2.
A: Thanks for the reviewer ’s suggestion, we have revised the content in the manuscript (p5)
